# Theoretical and Experimental Investigation of Surface Textures in Vibration-Assisted Micro Milling

**DOI:** 10.3390/mi15010139

**Published:** 2024-01-16

**Authors:** Bowen Song, Dawei Zhang, Xiubing Jing, Yingying Ren, Yun Chen, Huaizhong Li

**Affiliations:** 1Key Laboratory of Equipment Design and Manufacturing Technology, Tianjin University, Tianjin 300072, China; bowensong@tju.edu.cn (B.S.); medzhang@tju.edu.cn (D.Z.); ryy67853@163.com (Y.R.); 2Pen-Tung Sah Institute of Micro-Nano Science and Technology, Xiamen University, Xiamen 361102, China; yun.chen@xmu.edu.cn; 3School of Engineering & Built Environment, Gold Coast Campus, Griffith University, Southport, QLD 4222, Australia

**Keywords:** vibration-assisted machining, micro milling, surface texture machining

## Abstract

Vibration-assisted micro milling is a promising technique for fabricating engineered mi-cro-scaled surface textures. This paper presents a novel approach for theoretical modeling of three-dimensional (3D) surface textures produced by vibration-assisted micro milling. The proposed model considers the effects of tool edge geometry, minimum uncut chip thickness (MUCT), and material elastic recovery. The surface texture formation under different machining parameters is simulated and analyzed through mathematical modeling. Two typical surface morphologies can be generated: wave-type and fish scale-type textures, depending on the phase difference between tool paths. A 2-degrees-of-freedom (2-DOF) vibration stage is also developed to provide vibration along the feed and cross-feed directions during micro-milling process. Micro-milling experiments on copper were carried out to verify the ability to fabricate controlled surface textures using the vibration stage. The simulated and experimentally generated surfaces show good agreement in geometry and dimensions. This work provides an accurate analytical model for vibration-assisted micro-milling surface generation and demonstrates its feasibility for efficient, flexible texturing.

## 1. Introduction

The growing prevalence of engineered textured surfaces can be attributed to their ability to enhance the performance of component surfaces in various application areas, such as tribology [1], wettability [2], optics [3], bioengineering, and thermal properties [4,5]. The widespread incorporation of surface texturing across various industries has motivated the pursuit of sophisticated manufacturing techniques that can effectively create intricate surface morphologies. The repertoire of established surface texturing techniques includes focused ion beam machining [6,7], electron beam machining [8], lithography, laser ablation, and micro-rolling [9]. However, restrictions have been identified regarding economic feasibility, process efficiency, material compatibility, geometric complexity, and scalability. 

For many years, micromachining technology has dedicated significant efforts to investigating the microscopic scale material removal process [10,11]. Recently, vibration-assisted micro machining has arisen as a promising approach for accurate and flexible surface texturing [12,13,14]. Multiple advantages and improvements have been proved, such as reduced cutting forces [15], extended tool life, minimized burr formation, improved surface quality [16], and the ability to machine hardened metals and ceramics [17,18]. Vibration-assisted machining was initially designed to enhance the machinability of challenging materials, and later has proven to be adept at surface texturing. Among them, vibration-assisted micro milling has emerged as an efficient method for embedding surface textures on machined workpieces like aluminum alloy, copper alloy, and titanium alloy [19]. This technique overcomes limitations of existing approaches with the benefits of cost-effectiveness, high efficiency, and environmental friendliness. Engineered surface textures produced by vibration-assisted micro milling provide multifaceted performance enhancements, including reduced adhesion friction [20], smoothed lubricated sliding, upgraded wear resistance [21], controlled surface wettability [22], and tuned optical characteristics [23].

According to the vibration application types, vibration-assisted machining can generally be implemented in two ways: applying vibration to the tool and the workpiece. Since the cutter typically rotates at an extremely high speed during the micro-milling process, the second method of applying vibration to the workpiece is generally selected. Constructing a flexible stage is necessary to fix the workpiece to implement the method mentioned above [24]. Börner et al. [25] designed a cross-converter to deliver the vibration motion to the workpiece. Then, they discussed the application of ultrasonic vibration-assisted machining in end-milling processes to generate predefined microstructures. Ding et al. [26] described a 2-degrees-of-freedom (2-DOF) non-resonant flexible stage to fix the workpiece, and two piezoelectric actuators were symmetrically distributed to provide the vibration along the feed and cross-feed directions. 

In 3D surface texture modeling, Chen et al. [27,28] proposed a model based on homogenous matrices transformation, establishing the surface profile according to the tool tip’s shape. Lv et al. [29] developed a 3D vibration-assisted milling model for generating various structured surfaces by employing orthogonal spiral and multi-body kinematics theories. Yuan et al. [30] presented a vibration-assisted ball-end milling method for generating various surface textures using a non-resonant vibrator. Their model considered the tool trajectory in the vibration-assisted ball-end milling process, the quantitative relationships between the generated dimple geometric parameters, and the cutting and vibration condition. However, these models overlook the size effects inherent to the micro-cutting process, including the minimum uncut chip thickness (MUCT) phenomenon and material elastic recovery. This omission negatively impacts modeling precision.

To address the aforementioned issues, this paper introduces a novel method for forming surface texture in vibration-assisted micro milling, taking into account the influence of the tooltip geometry, MUCT, and material elastic recovery. The study aims to model the bottom side of the machined slot. Section 2 introduces the surface texture formation in the micro-milling process with different machining and vibration parameters. Section 3 presents the development of a novel 2-DOF vibration stage to realize the vibration during the milling process. Then, the vibration-assisted milling experiments are carried out to verify the proposed surface texture formation method in Section 4. Finally, the conclusion is made in Section 5.

## 2. Surface Texture Modeling

### 2.1. Tool Trajectory Modeling

To precisely establish the surface texture of vibration-assisted milling, the mathematical modeling of the conventional micro-milling surface should be first analyzed. Due to being most widely utilized in the micro-milling process, a micro-end-milling cutter with two flutes is employed on discussing for convenience.

Figure 1a shows a full slot micro end-milling operation for an end-mill with a two-flute cutter in the Cartesian coordinate system, in which, the *X*-axis and *Y*-axis represent the feed direction and cross-feed direction, respectively. *ω* is the spindle speed, *R* is the tool radius, and *φ* is the spindle phase angle. Figure 1b illustrates the tool trajectory of the micro-milling process, where the horizontal and vertical coordinates of points on the milling trajectory (*x*, *y*) can be calculated as follows:(1)$$\left\{ \begin{array}{l} x=ft+R\cos(\omega t+\pi \left(z_{i}-1\right)+\varphi )\\ y=R\sin(\omega t+\pi \left(z_{i}-1\right)+\varphi ) \end{array}\right.$$
where *z_i_* is the *i*th cutter flute, *f* is the feed rate, and *t* is the cutting time.

### 2.2. Three-Dimensional Micro-Milling Surface Modeling

Due to the micro dimension in the micro-milling process, the surface texture will be impacted significantly by the geometry of the tooltip, MUCT, and the material elastic recovery. Therefore, these effects should be considered in the surface texture modeling. The surface profile corresponding to the centerline of the slot floor is selected to discuss for convenience. Figure 2 illustrates the tooltip trajectory and machined surface profile at the centerline, with a close-up view of the bottom tool edge. In order to simplify the model, the tool edge is considered as an arc with a tool edge radius of *r_e_*, and the tool flank face is considered as a plane with a clearance angle of *γ*. Therefore, the profile height corresponding to the *x* coordinate can be defined as Equation (2):(2)$$Z_{e}=\left\{ \begin{array}{l} r_{e}-\sqrt{r_{e}^{2}-x^{2}},\;x>-r_{e}\sin\gamma \\ r_{e}-r_{e}\cos\gamma -\left[x+r_{e}\sin\gamma \right]\tan\gamma ,\;x\leq -r_{e}\sin\gamma \end{array}\right.$$

In micro-milling processing, an uncut chip thickness less than the MUCT results in no chip formation. To evaluate chip formation, a critical line for MUCT can be defined as follows [31]:(3)$${Ζ}_{h\min}=\{\begin{array}{c} -\sqrt{{\left(r_{e}-h_{\min}\right)}^{2}-x^{2}},x>-(r_{e}-h_{\min})\sin\gamma \\ -(r_{e}-h_{\min})\cos\gamma -[x+(r_{e}-h_{\min})\sin\gamma ]\cdot \tan\gamma ,x\leq -(r_{e}-h_{\min})\sin\gamma \end{array}$$

Based on the tool edge geometry considering the effect of MUCT and material elastic recovery, the generated surface profile can be depicted in Figure 2b. Where *f_z_*, *h*, *h*_min_, and *h_pre_* represent the feed per tooth, uncut chip thickness, MUCT, and the material elastic recovery height. *h_pre_* can be calculated as *h_pre_* = *h·P_e_*, where *P_e_* is the material elastic recovery ratio. The profile of the current flute (Z*_e_*_, *k*_) intersects point A and point D with that of the previous flute (Z*_e_*_,_ *_k_*_−1_) and that of the successive flute (Z*_e_*_, *k*+1_), respectively. The yellow area represents the cutting area. In the micro-milling process, the critical line for MUCT of current flute (Z*_h_*_min, *k*_) intersects with the profile of the previous flute (Z*_e_*_, *k*−1_) at point B, supposing that point B corresponding to the profile of the current flute (Z*_e_*_, *k*_) is recorded as point C, where BC is equal to the MUCT. On the left of point C, there is no chip formation due to the uncut chip thickness being less than the MUCT. The final profile in the BC zone is generated by elastic recovery of material after the current flute (Z*_e_*_, *k*_) passes the surface. The elastic recovery height could be calculated in our previous work according to Ref. [31]. For the right of point C, the surface material between point C and point D is removed by the successive flute (Z*_e_*_, *k*+1_), and the final profile generated (green line) coincides with the CD profile of the flute (Z*_e_*_, *k*_). This process of final profile in the AD zone is repeated, and the surface material can be removed entirely in the subsequent stage. Finally, the final surface profile is formed along the centerline of the slot floor until all flutes exit the workpiece, and the final surface profile can be expressed as follows:(4)$${Ζ}_{f}=\{\begin{array}{c} {Ζ}_{h\min}+(1-P_{e})({Ζ}_{e,k}-{Ζ}_{h\min})\\ {Ζ}_{e,k} \end{array}\begin{array}{c} h\leq h_{\min}\\ h>h_{\min} \end{array}$$

The feed per tooth significantly influences the machined surface profile. In a previous work, by comparing the feed per tooth and minimum uncut chip thickness in the milling process, two typical surface profiles were identified [31]. If *f_z_* > MUCT, a wave-type profile is formed; meanwhile, if *f_z_* < MUCT, a spike-type profile is produced. A set of micro-milling condition and cutting parameters are listed in Table 1. Under this condition with MUCT determined as 2.4 μm, two case studies for the surface profiles with the feed per tooth of 2 μm/tooth and 5 μm/tooth were conducted. A comparison of surface profiles with and without elastic recovery effects is illustrated in Figure 3a,b.

To derive the 3D surface texture, the 2D profile needs to be transferred based on the tool trajectory outlined in Equation (1). As shown in Figure 4, at a given time *t*, the tooltip has an angel *θ_t_* with respect to the *x* coordinate. The coordinate of a point on the tooltip can be designated as (*x_m_*, *y_m_*, *z_m_*). If the elastic recovery is not considered, the surface profile generated by the tooltip can be calculated using Equation (5):(5)$$\left\{ \begin{array}{l} \theta_{t}=\omega t+\pi \left(z_{i}-1\right)+\varphi \\ x_{m}=ft+(R+m)\cos\theta ,m<r_{e}\\ y_{m}=(R+m)\sin\theta ,m<r_{e}\\ z_{m}=Z_{e}(m),m<r_{e} \end{array}\right.$$
where *m* is equivalent with the *x* coordinate in Equation (2). By considering the elastic recovery as derived in Equations (3) and (4), the 3D surface texture could be established, as shown in Figure 5a,b.

### 2.3. Vibration-Assisted Micro-Milling Texture Modeling

According to the dimension of the vibration applied, vibration-assisted milling can be commonly classified into two types: 1-DOF vibration and 2-DOF vibration. In 1-DOF vibration-assisted micro-milling, vibration is delivered in either the feed or cross-feed direction, facilitating the movement of the workpiece in a single direction. In 2-DOF vibration-assisted milling, vibration co-occurs in the feed and cross-feed directions, causing an elliptical motion of the workpiece in a plane. When the 2-DOF vibration is applied to the workpiece, its trajectory could be calculated by Equation (6): (6)$$\left\{ \begin{array}{l} x_{w}=A\sin(2\pi f_{a}t+\theta_{x})\\ y_{w}=B\sin(2\pi f_{a}t+\theta_{y}) \end{array}\right.$$
where *A* and *B* are the vibration amplitudes, *f_a_* is the vibration frequency, and *θ_x_* and *θ_y_* are the phase angles in the x and y directions, respectively. As for a two-flute cutter, the relative displacement (*x*, *y*) of the tooltip to the workpiece in 2-DOF vibration-assisted milling can be obtained from Equations (1) and (7):(7)$$\left\{ \begin{array}{l} x=ft+R\cos\left[\omega t+\pi \left(z_{i}-1\right)+\varphi \right]+A\sin(2\pi f_{a}t+\theta_{x})\\ y=R\sin\left[\omega t+\pi \left(z_{i}-1\right)+\varphi \right]+B\sin(2\pi f_{a}t+\theta_{y}) \end{array}\right.$$

The essence of vibration-assisted milling is to apply high-frequency sinusoidal motion to the micro-milling trajectory, which changes the original micro-milling trajectory. Different phase angles result in differences in adjacent tooth trajectories, which can be combined into different surface morphologies. According to the micro-milling condition and vibration parameters presented in Table 2, it can be obtained that there are two typical surface morphologies. One is wave-type texture, which means that the surface morphology produced by adjacent flutes possesses the same phase angle, as shown in Figure 6a. Another is the fish-type texture, in which the phase difference between adjacent flutes is around 180 degrees, causing the surface morphology to interlock with each other, as shown in Figure 6b. The phase difference could be adjusted by changing the spindle speed and vibration frequencies. The conditions of the two cases to generate the above-mentioned two typical tool trajectories can be determined using Equation (8): (8)$$\left\{ \begin{array}{l} \frac{60f_{a}\cdot 2\pi }{nZ}=\left(2i+1\right)\pi ,i=1,2,3,\cdots \;\mathrm{Wave}\;\mathrm{type}\;\mathrm{texture}\;\mathrm{generation}\\ \frac{60f_{a}\cdot 2\pi }{nZ}=\left(2i\right)\pi ,i=1,2,3,\cdots \;\mathrm{Fish}\;\mathrm{type}\;\mathrm{texture}\;\mathrm{generation} \end{array}\right.$$
where *n* is the spindle speed, and *Z* is the number of flutes.

The 3D surface texture is obtained by using the mathematical method of conventional milling outlined in Section 2.2. For a given time *t*, the tooltip possesses an angel *θ_t_* with respect to the x coordinate. If the elastic recovery is not considered, but the 2-DOF vibration applied to the workpiece as in Equation (6) is included, the surface profile produced by the tooltip in Equation (3) can be updated to a new form as:(9)$$\left\{ \begin{array}{l} \theta_{t}=\omega t+\pi \left(z_{i}-1\right)+\varphi \\ x_{m}=ft+(R+m)\cos\theta +A\sin(2\pi f_{a}t+\theta_{x}),m<r_{e}\\ y_{m}=(R+m)\sin\theta +B\sin(2\pi f_{a}t+\theta_{y}),m<r_{e}\\ z_{m}=Z_{e}(m),m<r_{e} \end{array}\right.$$
where *m* is equivalent with the *x* coordinate in Equation (2). 

The flowchart for calculating the 3D surface texture is shown in Figure 7. By considering the elastic recovery as represented in Equations (3) and (4), the 3D surface textures of vibration-assisted milling under *f_z_* = 2 and 5 μm/tooth can be simulated, as shown in Figure 8a,b. According to the experiment conditions proposed in Equation (8), both wave and fish-type textures are derived, which proves the soundness of the analytical method.

## 3. Vibration-Stage Design and Optimization

### 3.1. Vibration-Stage Design

The basic structure of the proposed 2-DOF vibration-assisted platform is illustrated in Figure 9a, mainly consisting of a pair of piezoelectric actuators, a flexible stage, and a pedestal. To improve the assembly accuracy, the four corners of the flexible stage are positioned close to the pedestal. The schematic diagram of the flexible stage is shown in Figure 9b. To achieve the decoupling of two vibration directions, the whole structure is designed symmetrically, and a two-layer mechanism is utilized in the two-vibration direction. To further reduce the coupling effect, a novel kind of double-parallel flexure hinge is used as the outside framework of the stage due to the compression stiffness of the double-parallel flexure hinge being substantially higher than its rotational stiffness. The inner layer mechanism uses a single beam hinge to extend the mechanism’s vibration range. Additionally, circular-fillet hinges are employed throughout the construction. While increasing the motion precision, it could reduce the stress concentration.

### 3.2. Parameters Optimization

Flexible stages with large output displacement and high bandwidth are required to satisfy the demand for practical vibration-assisted milling. As the two most important parameters, the resonant frequency and compliance highly decide the bandwidth and output displacement of the flexible stage. Therefore, the resonant frequency and compliance are designed as optimization targets for structural parameters. The calculation method of compliance and the resonant frequency could be derived according to Ref. [32]. Based on the established theoretical model, parameter optimization of the 2-DOF stages is conducted to determine the detailed structural parameters. A typical suppression relationship exists between the two parameters, resulting in severe difficulty in the optimization process.

Based on the static and kinematic analysis results, the flexible stage’s performances are directly determined by the flexible hinges, including the inner and outer layer circular-fillet hinges. As shown in Figure 9b, the parameters of the hinges: fillet radiuses *r*_1_, *r*_2_, hinge lengths *l*_1_, *l*_2_, and widths *t*_1_, *t*_2_, are set as the optimization objections. The overall dimensions of the flexible stage are determined based on the milling machine’s working space [32]. Given the fixed dimensions of the fixed blocks and actuated blocks as illustrated in Figure 9b for the flexible stage, the parameters have the following relationship:(10)$$\begin{array}{l} 2r_{1}+t_{1}=3.5\;mm\\ 2r_{1}+l_{1}=19\;mm\\ 2r_{2}+l_{2}=4\;mm \end{array}$$

Therefore, the hinge widths of the outer-layer hinge *t*_1_, inner-layer hinge *t*_2_, and the fillet radius of the inner-layer hinge *r*_2_ can be modeled as the variable parameters in the parametric optimization process. The other parameters of the structure are given in Table 3. Due to the typical suppression relationship between the compliance and resonant frequency, the optimized objective function is defined as:(11)$$Q=k_{1}{\left(C-C_{1}\right)}^{2}+k_{2}{\left(k_{3}\left(f-f_{1}\right)\right)}^{2}$$
where *C*_1_ and *f*_1_ are the optimization objectives. *k*_1_, *k*_2_ are the weight factors and were set as *k*_1_ = 0.5 N/μm, *k*_2_ = 0.5 Hz^−1^. *k*_3_ is the balance factor, which can be defined as *k*_3_ = *C*_1_/*f*_1_. Al7075-T651 was adopted as the stage material, whose mechanical properties are shown in Table 4. According to the calculation results and experiment experience, the optimization objectives *C*_1_ and *f*_1_ were set as 0.1 μm/N and 3700 Hz. To guarantee the stage practicability of compliance and resonant frequency, the value of the three object parameters should be arranged in the range of 0.5–1.5 mm according to the experiment of the stage design. Subsequently, to ensure the integrity and accuracy of the database, each parameter was set as 0.5 mm, 1 mm, and 1.5 mm, resulting in 27 groups of combinations to be set as the initial data. The calculation results are shown in Table A1 in Appendix A. It can be observed that the 23rd group obtained the lowest value of *Q*, which means it has the closest data to the optimization objectives. In addition, the influence of parameters on compliance and frequency was analyzed based on the calculation results. Figure 10 shows the effect curve with one factor, while Figure 11 presents the influence contour with two factors. According to the graphs, it can be seen that with an increase in outer-layer hinge *t*_1_ and inner-layer hinge *t*_2_, the natural frequency gradually increases, and compliance presents a decreasing trend. At the same time, the fillet radius of the inner-layer hinge *r*_2_ has little impact on the two parameters. The influence of structural parameters on compliance and frequency can be ranked as *t*_2_, *t*_1_, *r*_2_.

The finite element method (FEM)-based simulation of the vibration stage was conducted by using the ANSYS Workbench 19.2. The meshing of the vibration stage was based on the hexahedron-dominated method, while the mesh refinement method was applied at the hinge connection. Considering the mode of the vibration stage in the prestressed state, the natural frequency and compliance of the vibration stage were obtained by constraining the four corner blocks.

The first- and second-order natural frequency shapes represent the translational motion of the vibration stage along the *Y*- and *X*-axes, respectively. Figure 12a shows that the axial displacement of the vibration stage is 4.5 µm for the case of the axial cutting force on the worktable at 100 N. Therefore, the axial stiffness and compliance of the stage are 22.22 N/µm and 0.045 µm/N, respectively, which satisfies the actual machining requirements. Figure 12b shows the stress distribution in the vibration stage. The findings reveal that the maximum stress value is 26.2 MPa, which is substantially lower than the yield strength of the material (455 MPa). Figure 12c shows the first resonance frequency of the stage, which corresponds to 3697.8 Hz. The difference between the optimized result and FEM simulation is within 10%, which indicates the excellent accuracy of the optimization model. 

## 4. Experimentation

### 4.1. Experiment Setup

To verify the surface texture models proposed in Section 2, a series of test experiments were designed and carried out on a precision three-axis micro-milling machine. The micro-milling machine was developed in-house using KXL/KYL/KZL 06050 C1 g series motorized stages from Suruga Seiki Company. The flexible stages were monolithically fabricated from Al7075-T651 by micro milling and wire electrical discharge machining methods. By setting the frequency and amplitude of the piezoelectrical actuator, an elliptic trajectory could be provided by the flexible stage and transmitted to the workpiece [32]. The experimental setup shown in Figure 13 mainly includes the flexible stage, micro-milling machine, input DAQ card, signal generator, and laser Doppler vibrometer. The flexible stages were driven by the two cylindrical encapsulated piezoelectric actuators (CoreMorrow, PSt150/10/20 VS15), with the preload devices built in to ensure the preloading force was identical in the two vibration directions. The driving signals were generated by the signal generator (Agilent 33500B) and further amplified by the piezoelectric-control system (Coremorrow’s E01.A2). Meanwhile, two laser Doppler vibrometers were fixed on the feed and vertical directions to monitor the flexible stage’s real-time output displacements. The displacement data were collected by the DAQ card (NI 9221) and sent to the LABVIEW 2016 software for analysis. The surface morphologies of the milling results were characterized by using an optical microscope (Leica DMi 8C) and a scanning electron microscope (SEM, ZEISS Sigma 300, Oberkochen, Germany).

The machining tests were carried out by using a 0.6 mm diameter uncoated two-flute micro end tool, while copper was chosen as the workpiece material for the convenience of observation. The experiment parameters are listed in Table 5. According to Equation (6), the parameters of tests 2 and 4 were within the condition of wave-type texture, while tests 3 and 5 were within the condition of fish-type texture. The cutting depth was all set as 50 μm.

### 4.2. Results and Discussion

Figure 14 shows the comparison between the machined surface textures and simulated textures, where the experimental results were observed by an optical microscope and a scanning electron microscope. Figure 14a shows the surface generated by the micro milling without vibration assistance, in which the circular tool marks are presented regularly. Figure 14b–e illustrates the surface texture of the surface machined by the vibration-assisted micro milling. Compared to the machining results of test 1, the surface texture shows a significant difference. Among them, the results of tests 2 and 4 in Figure 14b,d generated the wave-type texture, in which the peaks (valleys) of one tooth correspond to the peaks (valleys) of another tooth. The fish-type texture could be observed in the results of tests 3 and 5, as shown in Figure 14c,e. The peaks (valleys) of one tooth adjoin with the valleys (peaks) of another tooth precisely, as predicted in the simulated textures. The results show that vibration-assisted micro milling could be utilized as a new controllable and efficient method to generate certain surface textures.

The partially enlarged images of machined surfaces corresponding to test 2 and test 5 shown in Figure 15a,b were used to further study the influence of feed per tooth. The milling surface with a feed per tooth of 5 μm/tooth in test 5 exhibits a thick and wide texture compared to the result of test 2. In terms of size and geometry, the experimental results consistently align with the simulation results depicted in Figure 8. For a quantitative comparison, the surface profiles of the center line measured from the SEM images are shown in Figure 15c,d, together with the simulated surface profiles considering the elastic recovery using the proposed approach and the simulated surface profiles without considering the elastic recovery by utilizing the method from other references [27,28]. It is evident that the simulated surface profile considering elastic recovery exhibits significant consistency with the experimental results when compared with the simulation results without elastic recovery. This verifies the model’s capability to predict absolute dimensions. This supports the assertion that the vibration-assisted micro-milling technique is a flexible and efficient method for texturing, applicable to fabricating microfluidic devices, MEMS components, and engineered surfaces with customized optical, tribological, and biological properties. 

However, some subtle differences can be observed between the simulated and experimental surface profiles, which can be attributed to effects like tool wear, tool run-out and deflection, and plastic side flow, which are not included in the current model. In future work, the model can be further developed by incorporating more comprehensive tool-work interactions. Potential model refinements will be performed, including simulating variable chip thickness and runout effects for improved morphology prediction.

## 5. Conclusions

In this paper, a novel model was developed to predict the 3D surface textures in vibration-assisted micro milling by considering significant micro cutting factors. The simulation assessment and experimental verification demonstrated the capability for flexible and accurate texturing. The following key conclusions can be drawn:A precise analytical approach was proposed to model the surface generation process in vibration-assisted micro milling. This model incorporates critical size effects, including tool edge geometry, minimum chip thickness, and material elastic recovery, which enables reliable prediction of absolute texture dimensions.The modeling approach provides insights into texture formation mechanisms under various machining condition parameters. By controlling the spindle speed and vibration frequency, predictable wave-type and fish scale-type morphologies can be generated, as evidenced by both simulation and experiments.A 2-DOF flexible vibration stage was designed and fabricated. The micro-milling experiments demonstrated the feasibility of embedding controlled micro-scale surface patterns onto machined workpieces using this vibration-assisted method. Reasonable consistency was achieved between simulated and experimentally fabricated textures.

## Figures and Tables

**Figure 1 micromachines-15-00139-f001:**
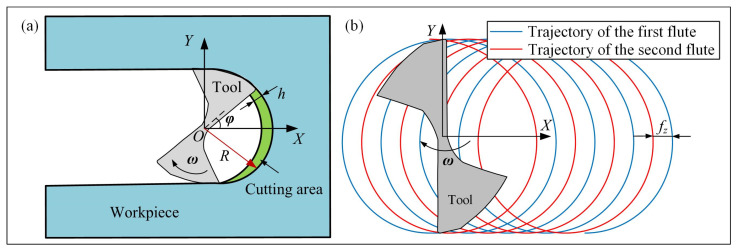
Micro-milling model: (**a**) micro end-milling operation, (**b**) tool trajectory.

**Figure 2 micromachines-15-00139-f002:**
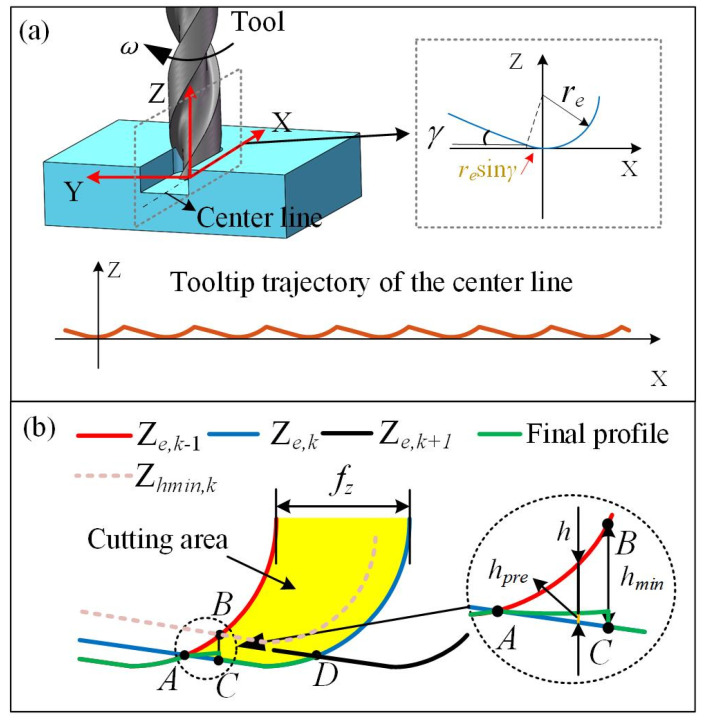
Micro-milling profile modeling: (**a**) the tooltip trajectory along the centerline, (**b**) machined surface profile along the centerline.

**Figure 3 micromachines-15-00139-f003:**
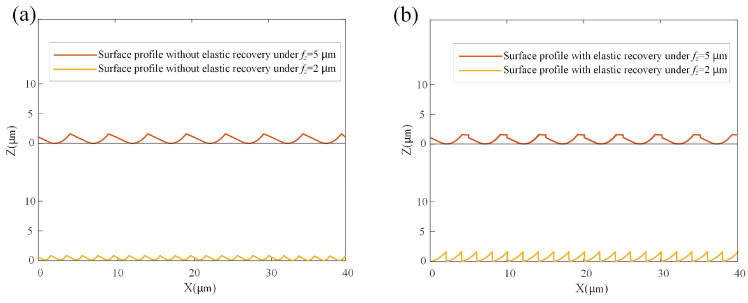
A 2D surface profile under different feeds per tooth (**a**) without considering elastic recovery, (**b**) considering elastic recovery.

**Figure 4 micromachines-15-00139-f004:**
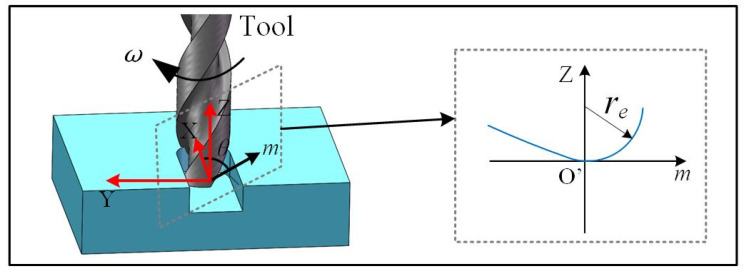
Three-dimensional surface profile modeling.

**Figure 5 micromachines-15-00139-f005:**
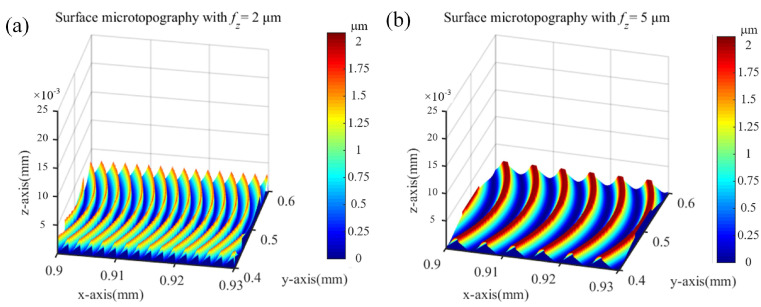
A 3D surface profile under different feeds per tooth (**a**) *f_z_* = 2 μm, (**b**) *f_z_* = 5 μm.

**Figure 6 micromachines-15-00139-f006:**
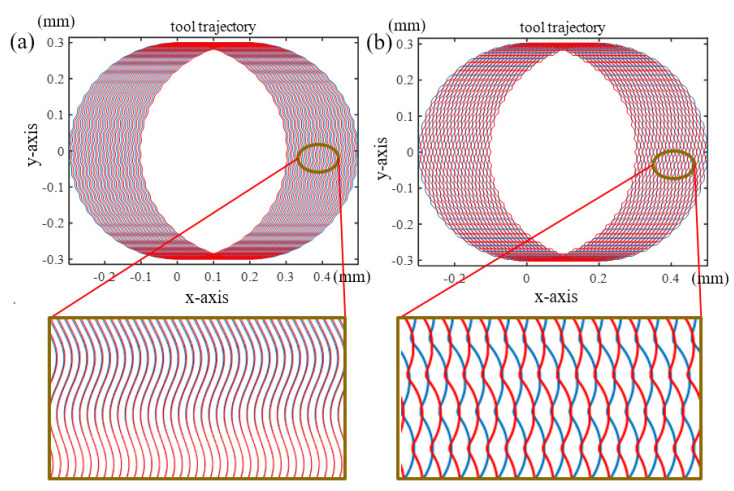
Two typical surface morphologies (**a**) wave-type trajectory, (**b**) fish-type tool trajectory.

**Figure 7 micromachines-15-00139-f007:**
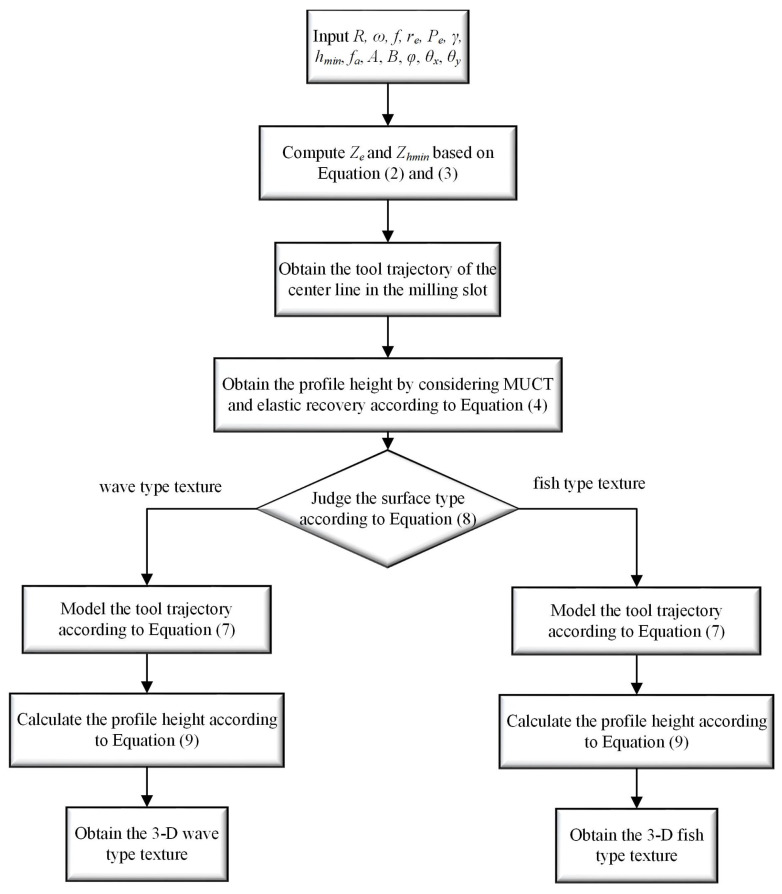
Flowchart to calculate the 3D surface texture.

**Figure 8 micromachines-15-00139-f008:**
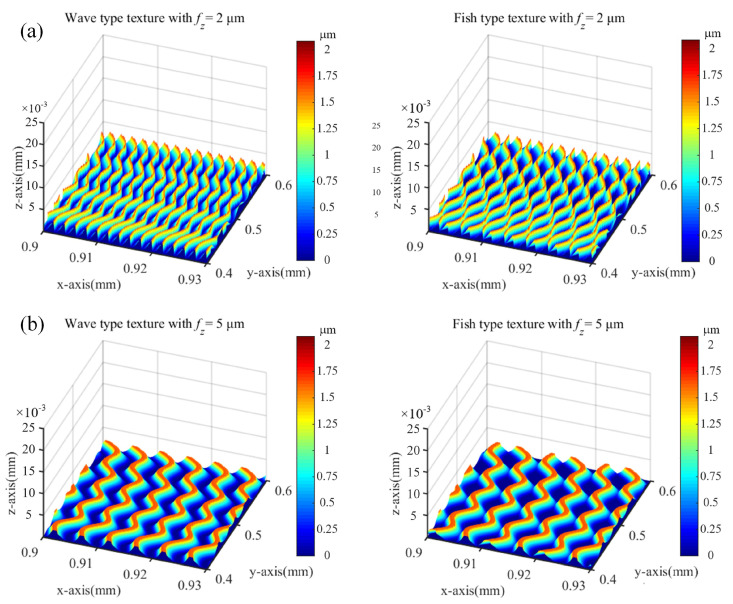
3D surface texture with vibration-assisted under different feed per tooth (**a**) surface texture with *f_z_* = 2 μm/tooth (**b**) surface texture with *f_z_* = 5 μm/tooth.

**Figure 9 micromachines-15-00139-f009:**
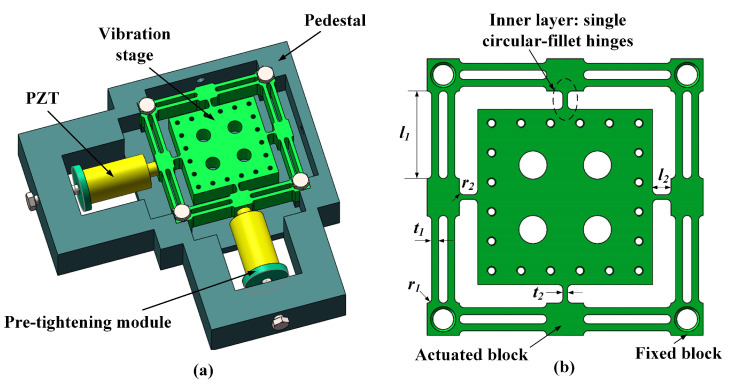
Mechanical design: (**a**) vibration-assisted stage, (**b**) flexible stage.

**Figure 10 micromachines-15-00139-f010:**
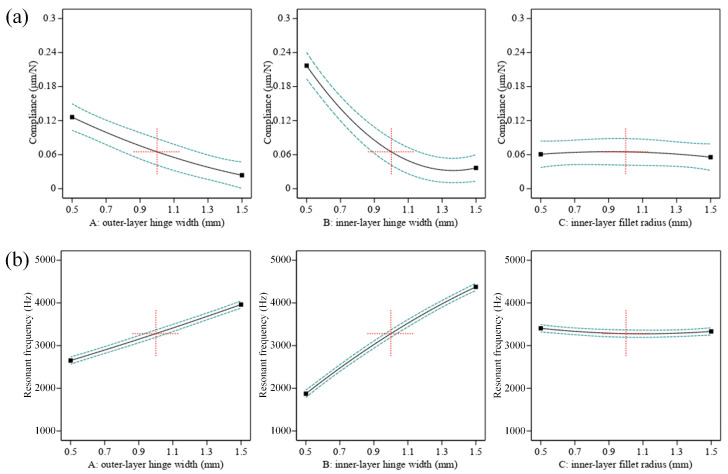
Influence of three parameters *t*_2_, *t*_1_, *r*_2_ on (**a**) compliance, (**b**) natural frequency.

**Figure 11 micromachines-15-00139-f011:**
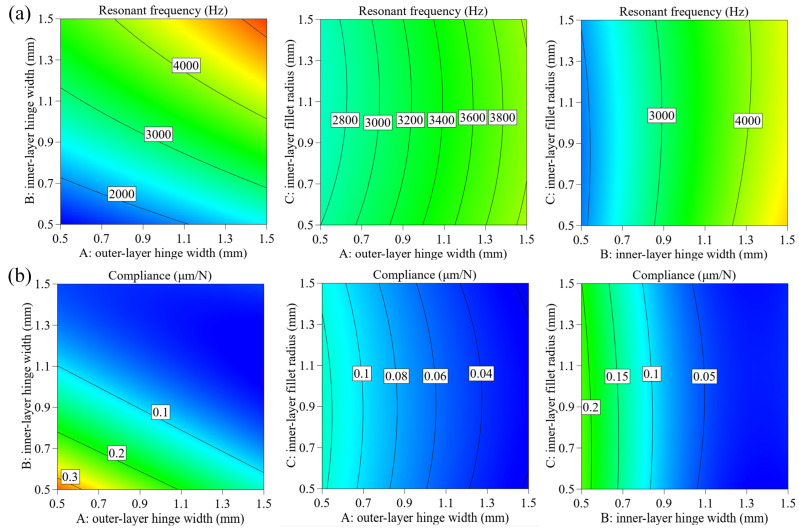
Contour of three parameters *t*_2_, *t*_1_, *r*_2_ on (**a**) compliance, (**b**) resonant frequency.

**Figure 12 micromachines-15-00139-f012:**
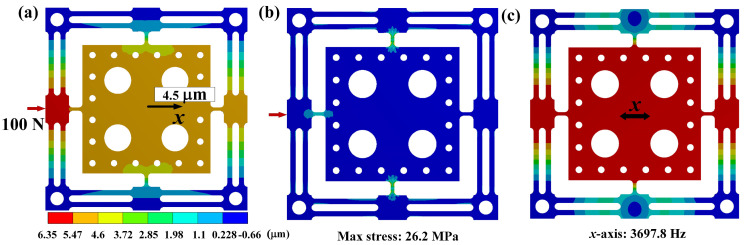
Finite simulation results: (**a**) static simulation, (**b**) stress distribution, (**c**) dynamic simulation.

**Figure 13 micromachines-15-00139-f013:**
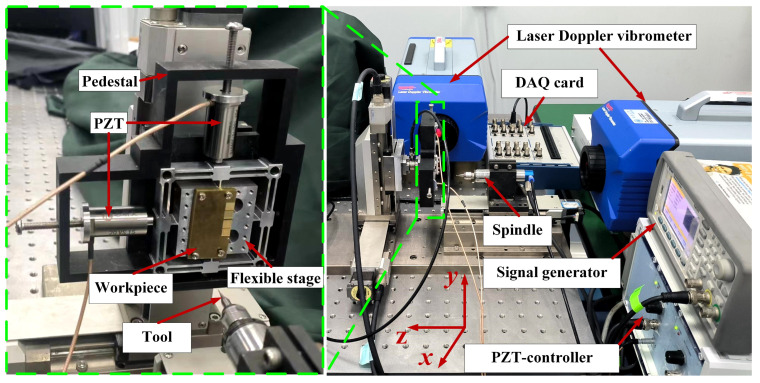
Experiment setup.

**Figure 14 micromachines-15-00139-f014:**
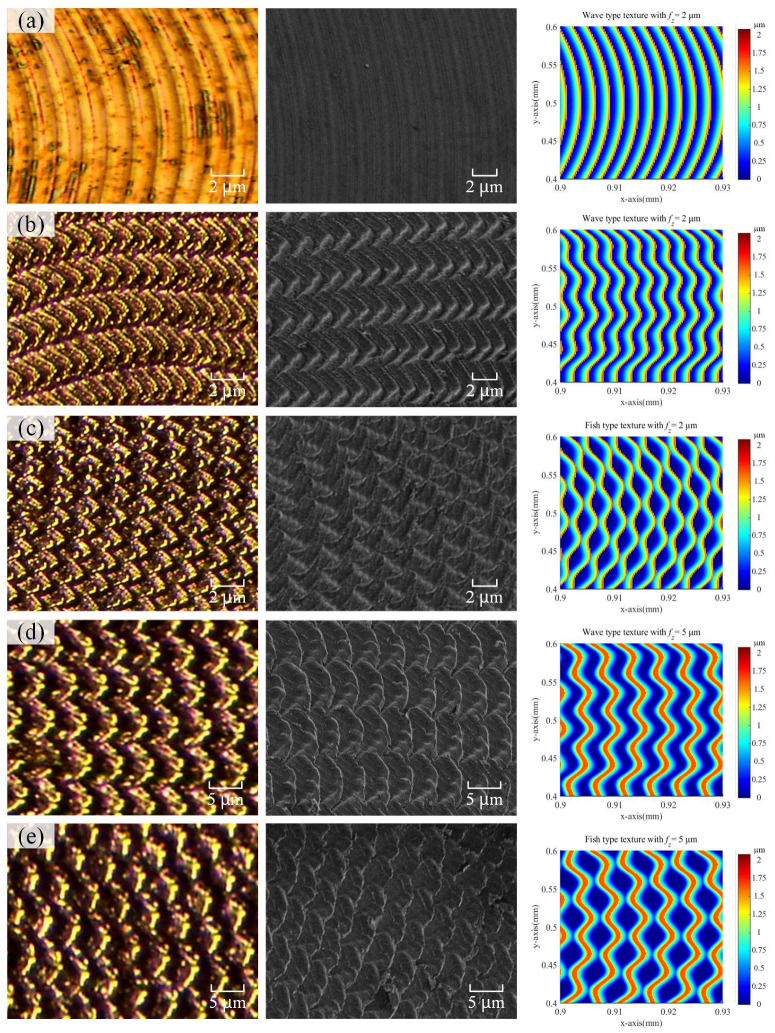
Machined surface textures observed by optical microscope and scanning electron microscope with conditions of test 1: (**a**); test 2: (**b**); test 3: (**c**); test 4: (**d**); test 5: (**e**).

**Figure 15 micromachines-15-00139-f015:**
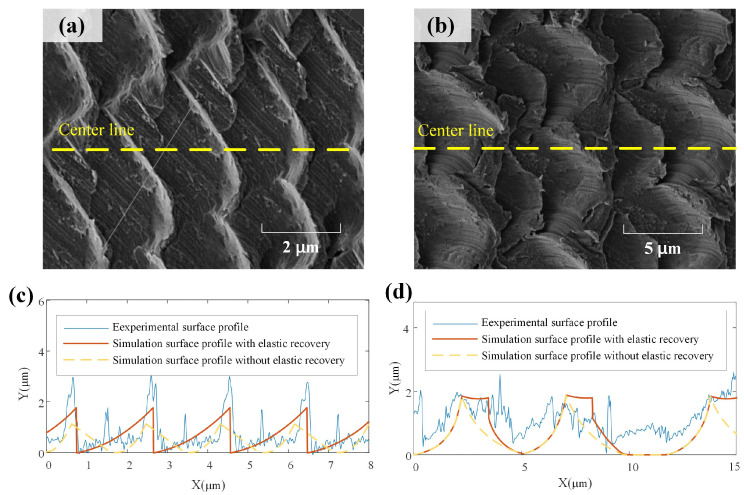
Partially enlarged surface profile: (**a**) surface profile of test 2, (**b**) surface profile of test 5, (**c**) center line profile of test 2, (**d**) center line profile of test 5.

**Table 1 micromachines-15-00139-t001:** Milling condition and cutting parameters.

Tool Radius (μm)	Spindle Speed (rpm)	Tool Edge Radius (μm)	MUCT (μm)	Material Elastic Recovery Ratio
600	5000	5	2.4	0.2

**Table 2 micromachines-15-00139-t002:** Simulation conditions.

No	Spindle Speed (rpm)	Vibration Frequency (Hz)	Vibration Amplitude (µm)	Phase Difference	Feed Per Tooth (µm)
Test a	5000	2417	2	90	2
Test b	5000	2500	2	90	2

**Table 3 micromachines-15-00139-t003:** Key parameters of the structure.

Parameters	*L* _1_	*L* _2_	*b*	*s* _1_	*s* _2_
Value (mm)	4	17	10	8	7

**Table 4 micromachines-15-00139-t004:** Mechanical properties of Al7075-T651.

Material	Density	Young’s Modulus	Poisson’s Ratio	Yield Strength	Tensile Strength
Al7075-T651	2.81 g/cm^3^	71 GPa	0.33	455 MPa	524 MPa

**Table 5 micromachines-15-00139-t005:** Experiment parameters.

No	Spindle Speed (rpm)	Vibration Frequency (Hz)	Vibration Amplitude (µm)	Feed Per Tooth (µm)
Test 1	5000	0	0	2
Test 2	5000	2500	2	2
Test 3	5000	2417	2	2
Test 4	5000	2500	2	5
Test 5	5000	2417	2	5

## Data Availability

Data are contained within the article.

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
