# Peer review of "Theoretical and Experimental Investigation of Surface Textures in Vibration-Assisted Micro Milling"

_micromachines, 2024, doi:10.3390/mi15010139_

Round 1

Reviewer 1 Report

Comments and Suggestions for Authors

Reviewer: The manuscript " Theoretical and experimental investigation of surface texture in vibration-assisted micro milling " proposed an improved surface texture model considering the effects of tool edge geometry, minimum uncut chip thickness (MUCT), and material elastic recovery. It needs minor revisions before publication. Some issues are listed below:

1. The English writing is suggested to be reviewed as some sentences are not easy to understand.

 2. The literature review in the introduction part is not well organized and should be presented with stronger logicality. Besides, most literature reviews should be present in the introduction part, instead of the results and discussion.

3. In Section 2.2, the author illustrates that two typical surface profiles are studied with the feed per tooth, while the conditions of the two types of surface should be given.

4. It is suggested to add some words to explain multiple formulas in modeling methods, in order to make the expression clearer.

5. Some mistakes should be checked carefully. Eg: "
By combining the tool trajectory and surface profile calculated in Equations (2) and (4), the 3-D surface textures of vibration-assisted under fz=2 and 5 μm could be simulated as shown in Figure 5a and b.", the unit of the feed per tooth (fz) should be μm/tooth.

6. The discussion on the advantages, disadvantages, and application scope of the model is suggested to be added.

Reviewer 2 Report

Comments and Suggestions for Authors

The paper seems interesting in the area of micromachining. However, some clarifications are required 

1. The obtained surface topography for the bottom of the wall sides?

2. In Eq. 2, it is necessary to model the surface roughness without applying an elastic rebound and then with applying an elastic rebound. Then this factor on roughness accuracy should be analyzed in all conditions (with and without vibration).  

3. Equation 2 should be clear and bring a better schematic diagram 

4. The implementation of the algorithm in programming software should be shown in a flowchart by bringing a flowchart addressing formulations and the time limit.  

5. for the vibration-assisted process in section 2.3, the equation for the calculation of the Z value of the 3D surface profile is missing. 

6. For better comparison of modeled and measured profiles, it is better to use some aspects of surface texture to quantify the results and show them in graphs.  

7. The application of textured AA7075-T651 should be emphasized to prove the significance of the work. 

Comments on the Quality of English Language

Quality of English is acceptable 

Reviewer 3 Report

Comments and Suggestions for Authors

1. The manuscript deals with the improvement of the model predicting surface textures in vibration-assisted micro milling based on the change of input technological parameters of the process. It describes how it is possible to produce surfaces of specific shapes through their change. This is a good choice for the production of parts with defined surface profiles, although in practice there are a large number of other machining methods, including post-treatment of their surface, through which these textures can be achieved more effectively.

2. I consider the topic of experimental research to be original, but the manuscript lacks an indication of the research gap in the context of the cited literature. Need to add.

3. The introduction of the manuscript contains a detailed description of known facts. However, it is necessary to focus on the description of problems in the field of surface textures in vibration-assisted micro milling, including an overview of the latest results achieved in the field by other researchers.

4. The manuscript is well written from a methodological point of view.

5. At the end of the manuscript, there is no discussion in which the authors indicate possible areas of application of the proposed solution to the research work. Also in the discussion, it is necessary to evaluate the achieved results of the experimental research in the context of the results achieved by other researchers in the field of surface textures in vibration-assisted micro milling. At the end of the manuscript, evaluate the qualitative or quantitative points of agreement or disagreement that the authors of the manuscript achieved when solving the experimental in comparison with the relevant references that were cited in the manuscript.

6. The references used in the manuscript are appropriate.

7. The manuscript is written at a level that corresponds to a research paper. After making minor adjustments, I recommend the manuscript entitled Theoretical and experimental investigation of surface textures in vibration-assisted micro milling for publication in the journal Micromachines.

Round 2

Reviewer 2 Report

Comments and Suggestions for Authors

the paper can be accepted for publication